# Intralayer and interlayer electron–phonon interactions in twisted graphene heterostructures

G.S.N. Eliel[1], M.V.O. Moutinho[2,3], A.C. Gadelha[1], A. Righi[1], L.C. Campos[1], H.B. Ribeiro[4], Po-Wen Chiu [5], K. Watanabe[6], T. Taniguchi[6], P. Puech[7], M. Paillet[8], T. Michel[8], P. Venezuela[3] & M.A. Pimenta[1]

The understanding of interactions between electrons and phonons in atomically thin heterostructures is crucial for the engineering of novel two-dimensional devices. Electron–phonon (el–ph) interactions in layered materials can occur involving electrons in the same layer or in different layers. Here we report on the possibility of distinguishing intralayer and interlayer el–ph interactions in samples of twisted bilayer graphene and of probing the intralayer process in graphene/h-BN by using Raman spectroscopy. In the intralayer process, the el–ph scattering occurs in a single graphene layer and the other layer (graphene or h-BN) imposes a periodic potential that backscatters the excited electron, whereas for the interlayer process the el–ph scattering occurs between states in the Dirac cones of adjacent graphene layers. Our methodology of using Raman spectroscopy to probe different types of el–ph interactions can be extended to study any kind of graphene-based heterostructure.

[1] Departamento de Física, Universidade Federal de Minas Gerais, UFMG, Belo Horizonte 30123-970, Brazil. [2] Núcleo Multidisciplinar de Pesquisas em Computação - NUMPEX-COMP, Campus Duque de Caxias, Universidade Federal do Rio de Janeiro, Duque de Caxias 25245-390 RJ, Brazil. [3] Instituto de Física, Universidade Federal Fluminense, UFF, Niterói 24210-346 RJ, Brazil. [4] MackGraphe - Graphene and Nanomaterials Research Center, Mackenzie Presbyterian University, São Paulo 01302-907, Brazil. [5] National Tsing Hua University, Hsinchu 30013, Taiwan. [6] National Institute for Materials Science, 1-1 Namiki, Tsukuba 305-0044, Japan. [7] CEMES/CNRS, University of Toulouse, 31055 Toulouse, France. [8] Laboratoire Charles Coulomb, CNRS, University of Montpellier, Montpellier 34095, France. These authors contributed equally: G. S. N. Eliel, M. V. O. Moutinho. Correspondence and requests for materials should be addressed to M.A.P. (email: mpimenta11@gmail.com)

**I**nterlayer electron–electron and electron–phonon (el–ph) scattering processes emerge from the coupling of atomic layers in two-dimensional (2D) heterostructures, and are essential for describing their physical properties and technological applications. The additional possibility of controlling the twisting angle $\theta$ between layers opens a fascinating route to achieve novel tunable quantum devices. For twisted bilayer graphene (TBG), the interaction between electrons of different layers generates van Hove singularities (vHs) in the density of electronic states (DOS), whose energies are $\theta$ dependent[1–3]. Electron–phonon coupling is also a fundamental interaction that affects a broad range of phenomena in condensed matter physics, such as electron mobility and thermal conductivity. In atomically thin heterostructures, the interaction can involve electrons in the same layer (intralayer el–ph interaction) or in adjacent layers (interlayer el–ph interaction). The interlayer el–ph interaction has been recently observed in $WSe_2$/h-BN heterostructures[4].

Here, we report the ability of Raman spectroscopy to probe and distinguish interlayer and intralayer el–ph interactions in graphene heterostructures. This is experimentally attained by tuning the energy of the excitation photon and observing the resonances of the Raman modes in different samples of TBG and graphene on the top of h-BN (gr/h-BN), with previously determined mismatch twisting angle $\theta$. Prominent new peaks are observed in the Raman spectra of TBG samples and come from phonons within the interior of the Brillouin zone (BZ) of graphene that are folded to the centre of the reduced Moiré pattern BZ. The frequencies of these phonons depend on the twisting angle $\theta$[5–11]. We show here that they can be activated either by the intralayer or the interlayer processes. The intralayer process was also observed in the Raman spectra of gr/h-BN samples and allows the experimental determination of their mismatch angle between the crystallographic axes of graphene and h-BN. In this case, the el–ph process occurs in a graphene monolayer and the h-BN surface imposes a periodic potential needed for the electron backscattering in the resonant Raman process. This effect is expected to be sensitive to the strength of the interaction between monolayer graphene and any other single layer or crystalline surface.

## Results

**Multiple-excitation Raman results in TBG using visible photons.** Experiments were first conducted in TBG flakes, that appear in an optical microscope as an external hexagon (first layer) and an internal one (with two layers), where the twisting angle $\theta$ can be determined from the optical images using the procedure reported in ref. [12] (see Supplementary Note 1 and Supplementary Figs. 1 and 2). Details of sample preparation can be found in ref. [13]. Figure 1a, b shows Raman spectra in two samples with $\theta = 6°$ and $13°$, recorded with the 2.18 and 2.41 eV laser lines, respectively. The vertical scale, $I_{TBG}/I_{SLG}$, corresponds to the ratio of the peak intensities in TBG and single-layer graphene (SLG). The ratio of $I_{TBG}/I_{SLG} = 1.8$ eV for the G-band intensity (around 1580 $cm^{-1}$) of the $\theta = 6°$ sample shown in Fig. 1a corresponds to

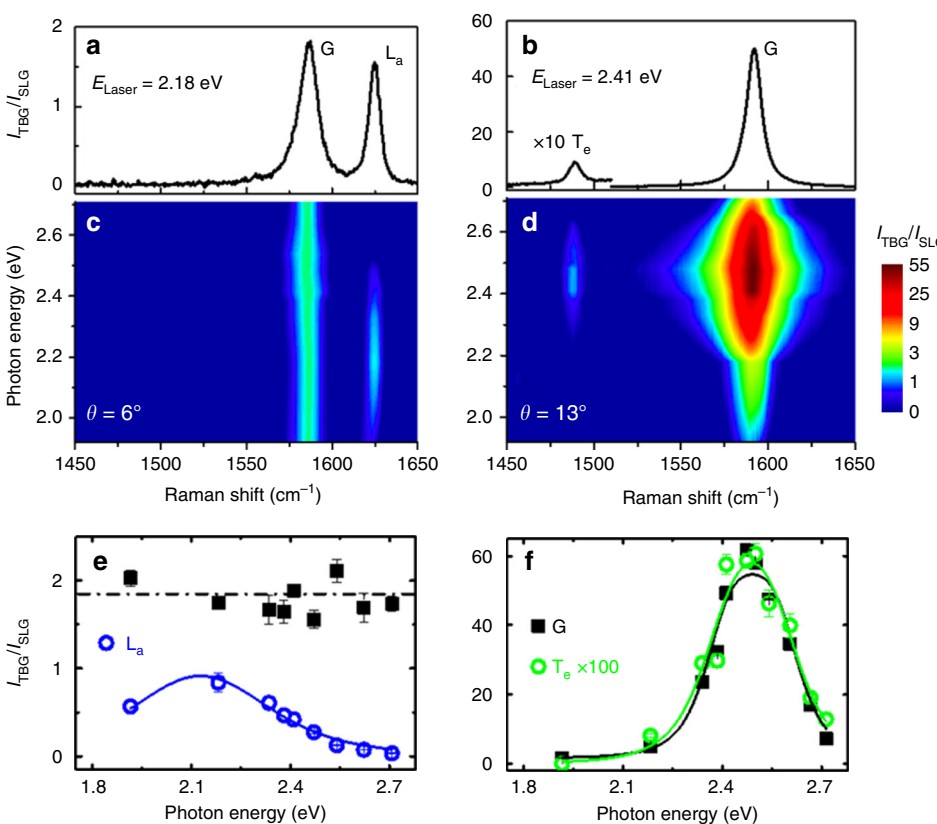

**Fig. 1** Raman results of TBG in the visible range. **a, b** Raman spectra in two samples of TBG with $\theta = 6$ and $13°$ recorded with the 2.18 and 2.41 eV laser lines, respectively. The vertical scale, $I_{TBG}/I_{SLG}$, corresponds to the ratio of the peak intensities of the Raman spectra in TBG and single-layer graphene (SLG). The peak around 1620 $cm^{-1}$ in **a** is called $L_a$, since it comes from the LO phonon branch and is activated by the intralayer electron–phonon scattering process, whereas the peak at 1480 $cm^{-1}$ in **b** is called $T_e$ since it comes from the TO phonon branch and is activated by the interlayer process. **c, d** Excitation Raman maps of the samples with $\theta = 6$ and $13°$ recorded with several laser lines with photon energies in the visible range (1.9–2.7 eV). **e** Raman excitation profile (REP) of the G band (black squares) and the $L_a$ peak (blue circles) of the sample with low twisting angle ($\theta = 6°$). **f** REP of the G band (black squares) and of the $T_e$ peak (green circles) of the sample with intermediate twisting angle ($\theta = 13°$). The $T_e$ peak intensity was multiplied by $\approx 100$ times for comparison with the G-band REP and in both **e, f**, the error bars represent the standard deviation

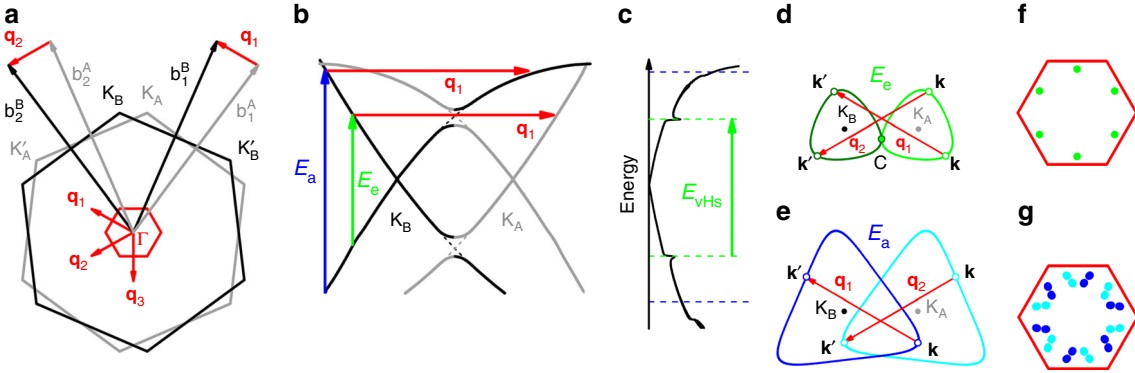

**Fig. 2** Intralayer and interlayer el–ph scattering processes. **a** The grey and black hexagons correspond to the Brillouin zones (BZs) of two graphene layers, denoted by A and B, twisted by the angle $\theta = 13.2°$. The small red hexagon represents the reduced BZ and the vectors $\mathbf{q}_1$, $\mathbf{q}_2$ and $\mathbf{q}_3$ correspond to the unit vectors of the Moiré reciprocal lattice. **b** Energy versus momentum diagram calculated for $\theta = 13.2°$. The grey (black) curves represent the Dirac cone of layer A (layer B). The vertical blue arrow represents the optical transition for the intralayer process, and the vertical green arrow represents the transition for the interlayer case. The horizontal red arrows represent the wave vector $\mathbf{q}_1$ of the phonon. **c** Density of electronic states (DOS) of the TBG and optical transition between vHs in the valence and conduction bands. **d** Interlayer el–ph process where a phonon with momentum $\hbar\mathbf{q}_1$ connects the states $\mathbf{k}$ and $\mathbf{k}'$. The light and dark green curves correspond to the equi-energies $E_e$ around $K_A$ and $K_B$. **e** Intralayer el–ph process where both states $\mathbf{k}$ and $\mathbf{k}'$ are in the equi-energies $E_a$ of the same layer (light and dark blue curves around $K_A$ and $K_B$, respectively). **f** Interlayer (green dots) and **g** intralayer (blue dots) electronic states $\mathbf{k}$ and $\mathbf{k}'$ represented in reduced BZ scheme

the observed value far from resonances[12]. However, for the $\theta = 13°$ sample shown in Fig. 1b, the G-band ratio of $I_{\rm TBG}/I_{\rm SLG}$ is more than 50. This huge increase of the G-band intensity has already been observed in previous Raman studies of TBGs[12, 14–16] and explained by the resonance of the incident photons with the transition between vHs in density of electronics states of a TBG.

In addition to the G band, we can observe in Fig. 1a, b the presence of extra peaks respectively above and below the G band, around 1620 and 1480 cm$^{-1}$. Extra peaks have been observed in many previous Raman studies of TBGs[5–11]. They arise from phonons of graphene with momentum $\hbar\mathbf{q}_M$, where $\mathbf{q}_M$ is a vector of the Moiré pattern reciprocal lattice, that are folded to the centre of the reduced BZ and become Raman active[5–11]. In 2010, Gupta et al.[5] proposed that the extra peaks were activated by double-resonance Raman (DRR) process where momentum conservation is provided by the potential of the Moiré reciprocal lattice. Carozo et al.[7] attributed the new peaks below and above the G-band position to the intervalley and intravalley DRR processes, and estimated the resonance energies as a function of $\theta$ for these two processes. The valleys considered in this work belong to the same graphene layer and, therefore, both cases correspond to an intralayer el–ph scattering process.

In 2013, Carozo et al.[10] and Wang et al.[11] reported measurements of TBG samples with intermediate twisting angles (13°–16°) using different laser lines. In both works, the appearance of new peaks in the range 1380–1450 cm$^{-1}$ was observed to occur in the same spectra where the G band was enhanced. This resonance behaviour could not be explained by the predictions of the DRR process involving the periodic potential of the Moiré[7], and revealed that a different el–ph process involving phonons with momentum $\hbar\mathbf{q}_M$ may exist. We will show below that the new phonons observed in these works[10, 11] are activated by the interlayer el–ph scattering process.

The extra peaks below and above the G-band position have been called in the literature as the R and R′ peaks[7, 10]. Since they can be activated either by the intralayer or the interlayer el–ph scattering process, we will adopt for these peak notation $A_\alpha$, where A = T or L refers to the branch (transverse optical, TO or longitudinal optical, LO) of the unfolded phonon and $\alpha$ refers to the el–ph scattering mechanism ($\alpha$ = a or e for intralayer and interlayer processes, respectively). It will be clear below that the

extra peaks above and below the G band shown in Fig. 1a, b might be called $L_a$ and $T_e$, respectively.

In order to explain the different resonance behaviour of the Raman peaks in TBGs, we first made multiple-excitation Raman measurements using many different laser lines in the visible range (see Methods section), which allowed us to obtain the accurate Raman excitation profile (REP) of Raman peaks in samples with small and intermediate twisting angles. Figure 1c, d shows the multiple-excitation Raman results for the samples with $\theta = 6$ and 13°, respectively. The vertical scale corresponds to the energy of the incident photon, and the peak intensities are represented by the colour bar on the right side. We can observe in Fig. 1c that the G-band ratio $I_{\rm TBG}/I_{\rm SLG}$ of the $\theta = 6°$ sample does not depend on the photon energy and is always around 1.8, an expected value far from resonance conditions[12]. On the other hand, the $L_a$ peak clearly exhibits a resonance behaviour and reaches the maximum intensity for photon energies around 2.2 eV, where it becomes as intense as the G band in SLG. Figure 1d shows that results for the $\theta = 13°$ sample exhibit a different behaviour. Now, both the G band and the $T_e$ peak exhibit a resonance behaviour and are enhanced in the same excitation energy range.

Figure 1e, f shows the REP of the G, $L_a$ and $T_e$ peaks, that is, the intensity of each peak as a function of the photon energy, for the $\theta = 6$ and 13° samples, respectively. Figure 1e shows that the REP of the $L_a$ peak exhibits maximum enhancement at approximately 2.2 eV and has a width $\gamma$ around 0.7 eV. This value of $\gamma$ agrees with the result in a previous ultraviolet (UV) Raman study of TBGs[8]. In the case of the $\theta = 13°$ sample shown in Fig. 1f, the G band and the $T_e$ peak exhibit very similar REPs, except for the fact that the intensity of the extra peak was multiplied by $\approx 100$ for comparison. The data in Fig. 1f were fitted by the expression of the Raman cross-section based on the third-order perturbation model[10], where the fitting parameters are the energies of the optical transitions among vHs, $E_{\rm vHs}$, and the width $\gamma$ of the REP. The values of the parameters that fit the experimental data in Fig. 1f are $E_{\rm vHs} = 2.37$ eV and $\gamma = 0.25$ eV. Notice that this value of $\gamma$ agrees with the width of the peaks in the optical absorption spectra of TBGs[17]. Several other samples with low (4°–6°) and intermediate (12°–16°) twisting angles were investigated using multiple-excitation Raman measurements, and results similar to those shown in Fig. 1 are presented in Supplementary Note 2 and Supplementary Figs. 3–5. The

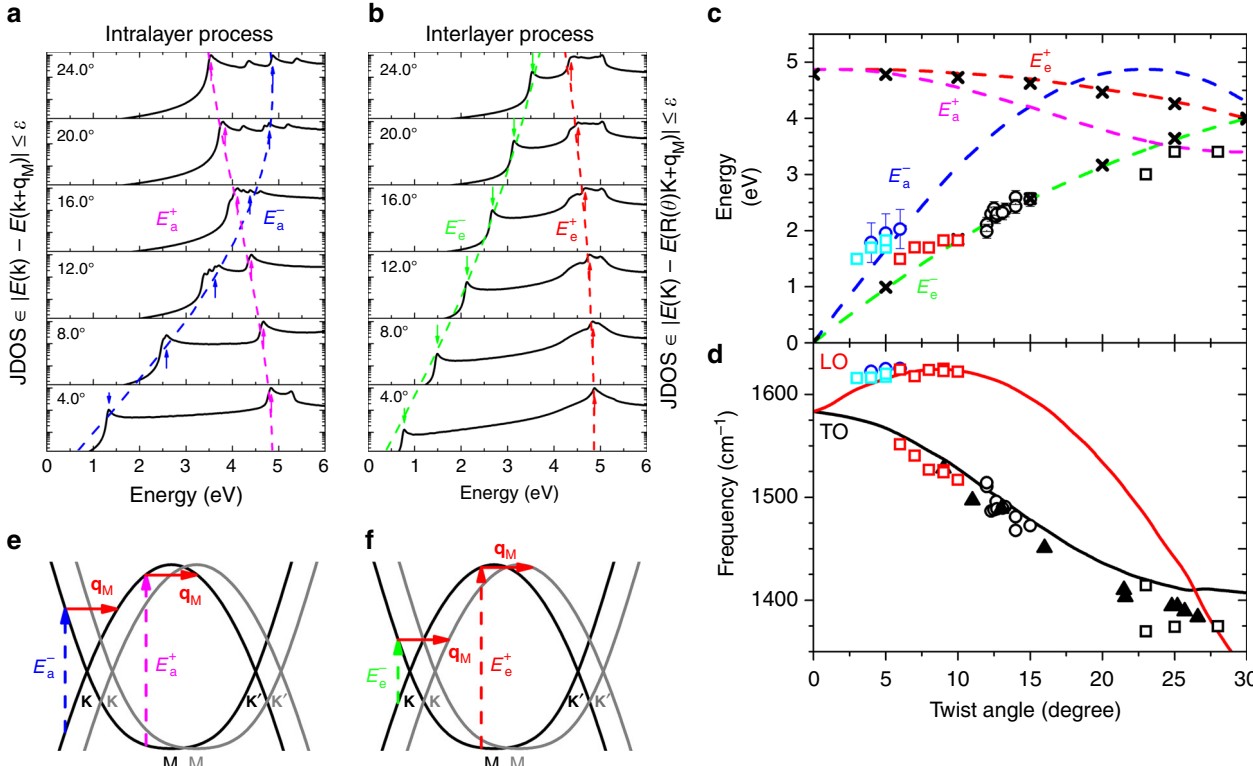

**Fig. 3** Resonance energies and phonon frequencies for the intralayer and interlayer el–ph processes. **a**, **b** Joint density of states that satisfy the intralayer and interlayer processes, respectively, for some twisting angles $\theta$. **c** The dashed blue, green, pink and red curves represent the calculated values of $E_a^-$, $E_e^-$, $E_a^+$ and $E_e^+$ as a function of the twisting angle $\theta$. The blue and black open circles correspond to the resonance energies of the extra peaks observed in the visible Raman spectra in samples with, respectively, small and intermediate angles. The light blue and red open squares correspond to the laser energies where the extra peaks observed in the Raman spectra in samples with $\theta$ in the ranges 3–5 and 6–10° have the maximum intensity. The black open squares correspond to the resonance energies of the extra peaks observed in the UV Raman spectra for samples with large twisting angles (22–28°). The black symbols × correspond to the energies of the peaks in the optical absorption spectra of TBG reported in ref. [17]. **d** The red and black curves represent the frequencies of the phonons with wave vector $\mathbf{q}_M$ from the TO and LO branches, respectively, as a function of $\theta$. The dark blue and black circles, and the light blue, red and black squares correspond to the frequencies of the extra peaks observed in the visible, IR and UV Raman spectra as described in **c**. The black triangles correspond to the results reported by Wang et al.[11]. **e**, **f** are schematic representation of the low- and high-energy intralayer and interlayer processes, respectively. The error bars in **c**, **d** represent the standard deviation

dependence of the values of $E_{vHs}$ and $\gamma$ for the G band on the twisting angle $\theta$ is shown in ref. [18].

**Intralayer and interlayer el–ph processes**. In order to understand the origin of the results presented above, we performed a theoretical simulation considering two different el–ph processes that take part in the Raman scattering. Details of the calculations are given in the Methods section. In one case, the excited electronic state from one layer is scattered by phonon with momentum $\hbar\mathbf{q}_M$ to another state in the same layer (intralayer process), whereas in the other case, the excited electron is scattered to a state of the other layer (interlayer process). We will show that these two mechanisms give rise to resonances at different energies.

Figure 2a shows the BZs of two graphene layers A and B twisted by $\theta = 13.2°$ and represented, respectively, by the grey and black hexagons. The vectors $\mathbf{q}_1$, $\mathbf{q}_2$ and $\mathbf{q}_3$ correspond to the unit vectors of the Moiré reciprocal lattice (we will consider in this work that $\mathbf{q}_M = \mathbf{q}_1$ or $\mathbf{q}_2$ or $\mathbf{q}_3$). The reduced BZ of the Moiré superlattice is shown by the small red hexagon in the centre of Fig. 2a. The intralayer and interlayer processes are represented in the energy versus momentum curves calculated for the $\theta = 13.2°$ TBG and shown in Fig. 2b. In the intralayer process, an incident photon with energy $E_a$ (represented by the vertical blue arrow)

creates an electron–hole pair, and a phonon with momentum $\hbar\mathbf{q}_M$ (represented by the horizontal red arrow) scatters the excited electron to another state in the same Dirac cone (layer B in this case). For simplicity, the change in energy between these two states is not considered in this figure since the phonon energy is much smaller than the energy of visible photons. In the interlayer process also represented in Fig. 2b, an incident photon represented by the green vertical arrow and with energy $E_e$ creates one electron–hole pair, but now the electron from one layer (layer B) is scattered to a state of the other layer (layer A) by a phonon with momentum $\hbar\mathbf{q}_M$. In both cases, the electron is then scattered back elastically to the first excited state by the Moiré potential, with wave vector $\mathbf{q}_M$, for electron–hole recombination and emission of the scattered photon.

Figure 2d illustrates the interlayer process in the graphene reciprocal space. The dark and light green curves correspond to the equi-energy curves $E_e$ around the Dirac points $K_B$ and $K_A$, respectively. An excited electron with momentum $\hbar\mathbf{k}$ from the Dirac cone of one layer (dark green curve) is scattered by a phonon with momentum $\hbar\mathbf{q}_M$ (red arrows) to a state with momentum $\hbar\mathbf{k}'$ in the Dirac cone of the other layer (light green curve). The lowest possible value of $E_e$ occurs when the equi-energies curves of the two layers tangentiate, as shown in Fig. 2d. In this situation, the anti-crossing between states of the Dirac cones gives rise to vHs in the density of states (DOS), shown in

Fig. 2c. For photons with energies below $E_e$, the difference $|\mathbf{k} - \mathbf{k}'|$ is always smaller than $|\mathbf{q}_M|$ and the interlayer condition cannot be satisfied. Therefore, the minimum energy for the interlayer scattering process, $E_e^-$, corresponds to the energy separation $E_{vHs}$ between the vHs in the valence and conduction bands of a TBG.

The intralayer el–ph scattering process is schematically represented in Fig. 2e. Now, the light and dark blue curves around $K_A$ and $K_B$, respectively, correspond to the curves of constant energy $E_a$. In this case, the excited states with momenta $\hbar\mathbf{k}$ and $\hbar\mathbf{k}'$ belong to the Dirac cones of the same layer (dark or light blue curves). The intralayer process is similar to the DRR mechanism that gives rise to the disorder-induced D and D′ bands in the Raman spectrum of graphene[19]. For the disorder-induced bands, the electron is scattered back to the initial excited state by a defect whereas, in the case of TBG, the backscattering is provided by a periodic potential of the Moiré pattern. This DRR process, where momentum conservation is provided by a vector $\mathbf{q}_M$ of the Moiré lattice, has been called previously as the umklapp DRR process[6].

So far, we discussed the two processes in the extended BZ scheme, where the activated phonons have finite $\mathbf{q}_M$. In the reduced BZ scheme, they are folded to the centre of the reduced BZ and have zero momentum. The first-order Raman process can be accomplished in this scheme since the two excited states $\mathbf{k}$ and $\mathbf{k}'$ in the extended BZ in Fig. 2d are folded to the same point in the reduced BZ, which are represented by the green dots in Fig. 2f. They are located at the saddle point in the electronic structure of TBG, near the M point in the reduced BZ, that gives rise to a vHs. The intralayer process can also be represented in the reduced BZ scheme. Figure 2g shows that the two-excited states $\mathbf{k}$ and $\mathbf{k}'$ in the extended BZ in Fig. 2e are folded to the blue dots within the reduced BZ. Different from the case of the interlayer process, which occurs near the M point of the reduced BZ, the intralayer process occurs for states at general positions within the interior of the reduced BZ.

In order to determine the resonance energies $E_a$ and $E_e$, we calculated the restricted density of joint electronic states that satisfy the intralayer and interlayer conditions. For the calculation, we divided the graphene BZ in a $2400 \times 2400$ $\mathbf{k}$-points grid and, for each twisted angle $\theta$, we stored the number of joint electronic states satisfying the restriction $|E^\alpha(\mathbf{k}) - E^\alpha(\mathbf{k}')| \le \varepsilon$ where the superscript $\alpha$ symbolises the valence or the conduction bands and $\varepsilon = 0.02$ eV is an arbitrary tolerance. For the intralayer case, we have $\mathbf{k}' = \mathbf{k} + \mathbf{q}_M$ because the two electronic states, $\mathbf{k}$ and $\mathbf{k}'$, are connected by $\mathbf{q}_M$ while for the interlayer process, $\mathbf{k}' = R(\theta) \mathbf{k} + \mathbf{q}_M$, where $R(\theta)$ is the rotation matrix that takes the Dirac cone of one layer into the Dirac cone of the other layer. Considering these two conditions, momentum conservation is always achieved in our model by a given pair of states, $\mathbf{k}$ and $\mathbf{k}'$, as shown in Fig. 2d, e.

Figure 3a, b shows the calculated joint density of states (JDOS) that satisfy the intralayer and interlayer resonance conditions, respectively. The results were smoothed by Lorentzian functions with 0.04 eV of FWHM and normalised by the maximum intensity for best visualisation of the peaks. In both cases, some peaks are observed in the restricted JDOS, and their energies depend on the twisting angle $\theta$. The position of the low-energy peaks increases, whereas the position of the high-energy peaks decreases with increasing values of $\theta$. These two maxima are associated with electronic transitions close to K and M points of the graphene, and will be called $E^-$ and $E^+$. For the intralayer process, the positions of $E_a^-$ and $E_a^+$ are indicated by the blue and purple arrows in Fig. 3a, respectively, and the positions of $E_e^-$ and $E_e^+$ for the interlayer process are marked in Fig. 3b by the green and red arrows. Notice in Fig. 3b that $E_e^-$ corresponds in fact to

the onset for the interlayer process, which occurs when the Dirac cones touch each other as shown in Fig. 2d, giving rise to vHs in the valence and conduction bands with energy separation $E_{vHs}$.

Figure 3c shows the calculated values of $E_a^-$, $E_e^-$, $E_a^+$ and $E_e^+$ as a function of the twisting angle $\theta$ represented by the dashed blue, green, pink and red curves, respectively. In Fig. 3e, f, we show the low- and high-energy processes for the intralayer and interlayer cases, respectively. Notice that, in the limit of small and large twisting angles, the low-energy transitions involve states near the K point of graphene, whereas the high-energy transitions involve states near the M point. The experimental values of the resonance energies obtained from the analysis of the REPs of the extra Raman peaks are also plotted in Fig. 3c. The blue open circles in Fig. 3d represent the resonance energies of the peaks in the range of 1600–1620 cm$^{-1}$ that are observed in samples with small angles (4°–6°). They nicely agree with the calculated $E_a$ versus $\theta$ results represented by the dashed blue curve. Details about the experimental values are shown in Supplementary Note 3 and Supplementary Figs. 6 and 7. The values of the resonance energies obtained from the REPs of the extra peaks in the range 1450–1550 cm$^{-1}$ for samples with intermediate twisting angles (12–16°) are plotted as open black circles in Fig. 3c. These points are close to the dashed green curve that represents $E_e^-$ as a function of $\theta$. The black symbols × in Fig. 3c represent the experimental values of optical transition energies $E_{vHs}$ measured directly by optical conductivity[17]. It is interesting to note that, despite the fact that our model does not consider electronic coupling between the layers that opens a minigap which splits the valence and the conduction bands in TBL, the agreement between the minimum possible value $E_e^-$ shown in Fig. 3a and the experimental values of the optical transitions[17] is remarkable.

The red and black curves in Fig. 3d display, respectively, the calculated frequencies of LO and TO phonons with momenta $\hbar\mathbf{q}_M$ as a function of $\theta$. The frequencies of the extra peaks of samples with small (intermediate) values of $\theta$ are plotted as open blue (black) circles in Fig. 3d. By comparing our experimental results with the calculated results of the resonance energies and phonon frequencies shown in Fig. 3c, d, we conclude that using visible photons, the Raman peaks below the G band come from TO phonons and are activated by the interlayer process, whereas the peaks above the G band come from the LO phonon branch and are activated by the intralayer process.

In principle, the intralayer and interlayer el–ph processes can also activate TO and LO phonons, respectively, and give rise to $T_a$ and $L_e$ peaks for samples with intermediate angles. However, $T_a$ and $L_e$ were not observed in our multiple-excitation experiments using visible photons. The lack of observation of $L_e$ can be due to the huge enhancement of the G band. Since the position of the $L_e$ peak is very close to the G-band position in samples with intermediate twisting angles, it is possibly masked by the G-band enhancement. For samples with $\theta$ around 10° and measured using the 1.96 eV laser line, Campos-Delgado et al.[9] reported the observation of a peak at 1622 cm$^{-1}$, that might be assigned to the $L_e$. The absence of $T_a$ can be ascribed to the very weak cross-section of TO phonons activated by the intralayer process. Electron–phonon matrix elements calculations would be necessary to better clarify this issue.

## Raman spectra of TBG using IR and UV photons

The calculated results presented in Fig. 3c show that, for TBG samples with small twisting angles, both the intralayer and the interlayer processes are expected to be observed using excitation energies below 1.9 eV. This prediction was indeed observed in our Raman results performed in many different samples with $\theta$ in the range of 2–9°, using three laser lines with energies 1.49, 1.70 and 1.82 eV (see

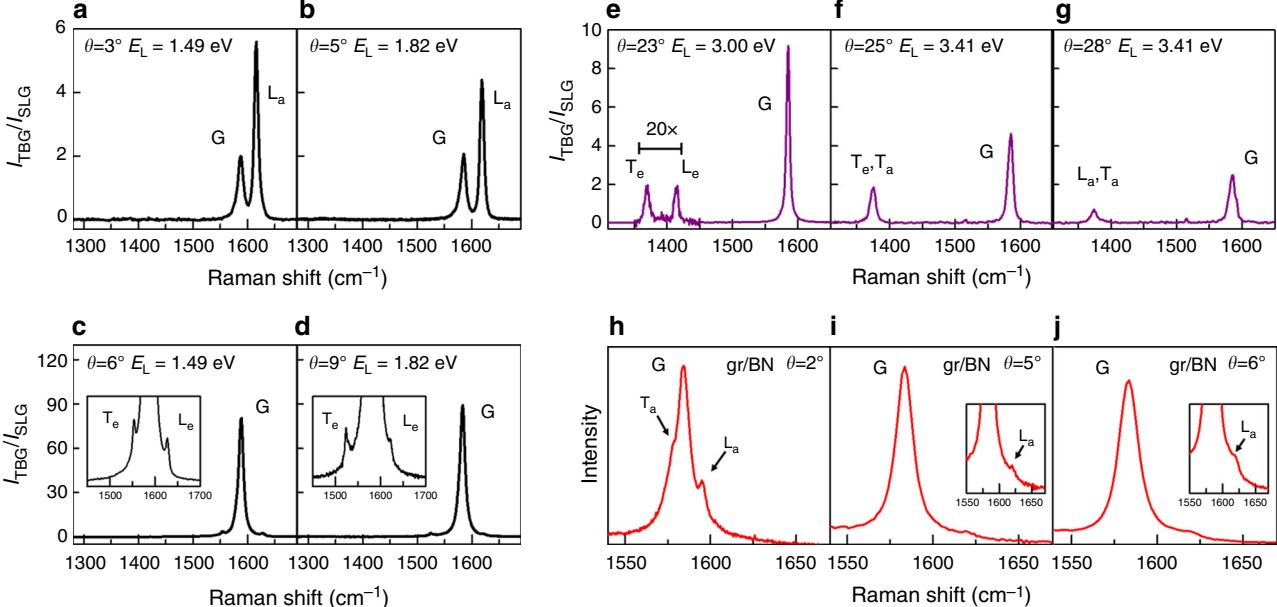

**Fig. 4** Raman results in TBGs using IR, visible and UV radiation and results in gr/h-BN samples. **a–d** Raman spectra in four TBG samples with $\theta = 3$, 5, 6 and 9°, recorded with the 1.49 or 1.82 excitation energies, as indicated in each figure. **e–g** UV Raman spectra of three TBG samples $\theta = 23$, 25 and 28°, recorded with the 3.00 or 3.41 excitation energies. **h–j** Raman spectra in three different samples of graphene on the top of h-BN, with twisting angles $\theta = 2$, 5 and 6°, recorded with the 1.96 eV excitation energy

Methods section, Supplementary Note 4 and Supplementary Fig. 8). The Raman spectra for samples with $\theta = 3$, 5, 6 and 9° are presented in Fig. 4a–d. In the spectra of the samples with $\theta = 3$ and 5°, shown in Fig. 4a, b, the ratio of $I_{TBG}/I_{SLG}$ of the G band around 2 reveals that the LO phonon is activated by the intralayer el–ph process. Interestingly, the $L_a$ peak is stronger than the G band, suggesting that the el–ph coupling for the LO phonon increases with decreasing $\theta$, when the wave vector $q_M$ tends to the centre $\Gamma$ of the BZ. This effect is shown in the Supplementary Note 5 and Supplementary Fig. 9 that plots the relative intensities of $T_a$, $T_e$, $L_a$ and $L_e$ as a function of the twisting angle $\theta$. In contrast, the interlayer el–ph process occurs for the TBG samples with $\theta = 6$ and 9° shown in Fig. 4c, d, where a huge enhancement $I_{TBG}/I_{SLG} \approx 80$ is observed. In the tails of the G band, we can observe two extra peaks below and above the G band that are assigned, respectively, as the $T_e$ and $L_e$ peaks. The new result here is the signature of a LO phonon activated by the interlayer el–ph process ($L_e$ peak) that was not observed in the results obtained with visible photons and shown in Fig. 1. This result can also be explained by the increase of el–ph coupling for the LO phonons when $q_M$ tends to zero. The laser energies where the extra peaks observed in the infrared (IR) Raman spectra in samples with $\theta$ in the ranges of 3–5 and 6–9° have the maximum intensity are shown, respectively, by the light blue and red open squares in Fig. 3c. The frequencies of the corresponding peaks are plotted in Fig. 3d and are in nice agreement with the calculated values.

The theoretical predictions in Fig. 3c for samples with large twisting angles $\theta$ show that extra peaks are expected to appear in the UV Raman spectra (for excitation energies above 3 eV). Figure 3d shows that the activated phonons are close to the K point of the SLG, and come from both the TO and LO branches, with frequencies in the range of 1350–1420 cm$^{-1}$. In the pioneer Raman study of graphene folded onto itself, Gupta et al.[5] observed nondispersive extra peaks in the range of 1370–1395 cm$^{-1}$, and showed that they were enhanced in the UV Raman spectrum ($E_{laser} = 3.41$ eV). Righi et al.[6] studied the UV Raman spectra of TBGs and observed several extra peaks in the range of 1370–1420 cm$^{-1}$, but the twisting angles $\theta$ were not determined

in this work. In both studies[5, 6] the appearance of new peaks was not accompanied by the enhancement of the G band, as expected for an intralayer el–ph process. In contrast, the UV data of Wang et al.[11] in TBG samples with large values of $\theta$ show the activation of new peaks in spectra where the G band also exhibits an enhancement of 5–20 times, indicating thus an interlayer el–ph scattering. The laser energies where the extra peaks of the samples with large values of $\theta$ in the range of 22–28° exhibit the largest intensity and the frequencies of these peaks observed in the UV Raman spectra are represented by black open squares in Fig. 3c, d, respectively. The results for the phonon frequencies reported by Wang et al.[11] are also plotted in Fig. 3d, and represented by black triangles.

Figure 4e–g shows the UV Raman spectra of three samples with $\theta = 23$, 25 and 28°, performed with the 3.00 and 3.41 eV excitation energies (details are described in the Methods section). The vertical scale gives the ratio of $I_{TBG}/I_{SLG}$. We can see in the spectra new peaks with frequencies in range of 1370–1420 cm$^{-1}$, as predicted in Fig. 3c, d. Notice in Fig. 3c that for large twisting angles, both the intralayer ($E_a^+$) and interlayer ($E_e^-$) resonances occur in the same energy range (3–4 eV). In principle, we can distinguish these two processes by observing the enhancement of the G band of TBG, which is a signature of the interlayer process.

In the spectrum of the $\theta = 23$° sample recorded with the 3.00 eV line and shown in Fig. 4e, the ratio of $I_{TBG}/I_{SLG}$ for the G band is $\approx 9$, suggesting that the two peaks around 1380 and 1410 cm$^{-1}$ come from phonons of the TO and LO branches, respectively, and are activated by the interlayer el–ph process ($T_e$ and $L_e$ peaks). On the other hand, in the spectrum of $\theta = 28$° sample recorded with the 3.41 eV line and shown in Fig. 4g, the ratio of $I_{TBG}/I_{SLG} \approx 2$ for the G band reveals the activation by the intralayer el–ph process. We thus assign the extra feature around 1370 cm$^{-1}$ as a $L_a$ peak, although we cannot rule out the possibility of being a $T_a$ peak, since, as shown in Fig. 3d, the LO and TO phonon branches are close to each other in the range of large $\theta$. In Fig. 4f, the ratio of $I_{TBG}/I_{SLG}$ for the G band is around 4 and only one extra peak is observed. In this case, possibly both processes are occurring simultaneously. The peak position around

$1380 \text{ cm}^{-1}$ suggests that it comes from a phonon of the TO branch, and it can be thus assigned as a $T_a/T_e$ peak. The laser energies where the extra peaks shown in Fig. 4e–g exhibit the largest intensity and their positions are also plotted in Fig. 3c, d, respectively, and compared with the calculated TO or LO branches near the K point of graphene. The agreement between the experimental and theoretical values of the phonon frequencies is not as good as in the case of the results in the visible range. This is due to the shift in the calculated phonon frequencies described in the Methods section.

**Intralayer process in any graphene heterostructure**. As discussed above, the intralayer el–ph process occurs in one graphene layer, and the second layer only imposes a periodic potential needed for momentum conservation in the DRR process. Therefore, extra peaks are also expected to appear in the Raman spectra of graphene deposited on top of any atomically flat substrate, assuming that the interaction graphene/substrate is strong enough to scatter electrons. In a recent study of graphene on the top of h-BN, Eckmann et al.[20] reported the observation of weak extra peaks both below and above positions of the G band in graphene, and suggested that they arise from the graphene/h-BN interaction.

In order to check this assumption, we prepared graphene/BN samples by transferring SLG to the top of a h-BN crystal. From the analysis of optical images, where the crystallographic edges of graphene and BN are evidenced, we can estimate the twisting angle $\theta$ (see Supplementary Fig. 2). Raman measurements were performed in graphene/h-BN samples with different twisting angles, especially in the range of small $\theta$ where the intralayer process occurs for photons in the visible range. Figure 4h–j shows the Raman spectra in three samples of graphene/h-BN, with $\theta = 2$, 5 and 6°, in the range of $1550–1650 \text{ cm}^{-1}$. In the three cases, we can observe very weak peaks above the G-band frequency, assigned as $L_a$ peaks. In the case of the $\theta = 2°$ sample, we can also observe in Fig. 4h a sharp peak below the G band, and this result is thus an experimental manifestation of the $T_a$ process. The positions of the extra peaks in graphene/BN are approximately the same of TBG graphene, except for the case of samples with very small twisting angles ($\theta < 2°$)[20]. They are much less intense than the extra peaks in bilayer graphene showing that the imposed potential of h-BN on graphene is much weaker than graphene–graphene interaction. Results shown in Fig. 4h–j, thus, demonstrate that extra peaks enhanced by the intralayer el–ph process in TBG occur whenever graphene interacts with an atomically flat crystalline surface that imposes a periodic potential for the electrons of graphene.

## Discussion

We report in this work the observation of intralayer and interlayer el–ph interactions in twisted graphene heterostructures with different mismatch twisting angles by Raman spectroscopy. Measurements performed with many different laser excitation energies allowed us to conclude that phonons of graphene with momenta $\mathbf{q}_M$ can be activated by two different resonant el–ph processes: the interlayer process, where the electron scattering occurs between the Dirac cones of different graphene layers, and the intralayer process which occurs in a single graphene layer, and the other layer or substrate only imposes a periodic potential that activates the phonons in the Raman spectrum. The observation of new peaks in the Raman spectrum of SLG on the top of h-BN crystals proves that the intralayer el–ph process occurs not only in TBG, but also whenever graphene is in contact with any periodic layer or substrate. The intensity of these extra peaks is expected to provide the strength of the interaction between

monolayer graphene and the substrate. On the other hand, the activation of new phonons by the interlayer el–ph process reported in many previous studies in the literature occurs only in TBGs.

This work highlights the importance of understanding fundamental el–ph interactions in heterostructures of 2D materials, which affect thermal and transport properties in these systems. The possibility of distinguishing intralayer and interlayer el–ph interactions in bilayer graphene and of studying the interaction of monolayer graphene with a crystalline substrate using light scattering allow new ways to design devices applications of 2D heterostructures and is crucial for the engineering of new devices.

## Methods

**Raman measurements**. The Raman measurements of the TBG samples were performed using many different laser lines, in the near IR, visible and UV ranges, for samples with different twisting angle $\theta$. Experiments in the visible range were performed in a DILOR XY triple-monochromator spectrometer equipped with a $N_2$-cooled charge-couple device detector, a 1800 g/mm diffraction grating, and using an Ar/Kr with 12 laser lines in the visible range. The NIR Raman measurements were performed on a home-made setup, including an iHR-550 Horiba spectrometer equipped with a liquid-nitrogen-cooled silicon CCD detector. A tunable CW Ti:sapphire laser filtered using tunable laser line filters[21] was used for excitation. The scattered light was collected through a ×100 objective (NA = 0.95) using a backscattering configuration. The 3.41 eV Raman spectra were obtained using a Dilor UV Raman spectrometer equipped with a cooled CCD and an argon-ion laser. The laser power on the sample was kept lower than 1 mW with ×40 objective to avoid heating. The Raman spectra at 3.00 eV were obtained using a Jobin-Yvon-Horiba T64000 Raman spectrometer equipped with a cooled CCD with a krypton-ion laser. The laser power on the sample was limited to 0.5 mW with ×100 objective. In the case of G/h-BN heterostructures, the measurements were performed in Witec Alpha 300R system using 633 nm as a pump laser.

**Sample fabrication**. The TBG samples were grown by CVD technique using methane (99.99%) on polycrystalline Cu foils[13]. The graphene layers were first covered by a thin layer of polycarbonate, followed by etching in HCl aqueous solution to remove the Cu in the transfer process. The polycarbonate film with attached graphene was then transferred onto different substrates: a 300 nm SiO₂/Si, 90 nm SiO₂/Si and fused silica. Finally, the polycarbonate film was removed using chloroform. The gr/BN samples were prepared by mechanical exfoliation of graphene and transference to a h-BN substrate.

**Theoretical model**. The electronic and phonon structures were obtained by folding the SLG calculation results. For SLG, we follow the calculation procedure given in ref. [22]. The electronic structure calculations are based on a tight-binding approach in which the parameters are fitted to reproduce density functional theory (DFT) calculations with many-body corrections, while the phonon dispersion was obtained from many-body corrected DFT calculations. The many-body corrections change the phonon slope of the highest optical branch near K (with respect to DFT), reproducing the Kohn anomaly and providing a much better agreement with inelastic X-ray scattering measurements for graphite[23]. These curves describe accurately the D band of graphene when a red shift of $40 \text{ cm}^{-1}$ is applied for phonons close to the K point. However, in order to reproduce the phonons close to the graphene G band, we blue shifted all frequencies by $20 \text{ cm}^{-1}$. Even though the calculations were done only by folding the SLG results, the physics of the intralayer and interlayer processes in TBG is nicely captured.

Concerning the electronic properties, it has been shown that interlayer interactions perturb the bands from each layer producing minigaps in the conduction and valence band of TBG with same energy. However, the allowed transitions do not depend on the size of the minigap, and therefore the folding approach is supposed to give good results concerning the Raman processes investigated here[17]. As far as phonon properties are concerned, Cocemasov et al.[24] have shown that the weak van der Waals interactions between layers in TBG do not alter significantly the frequencies of the branches in the range considered here. Although the folding method provides a good description of the phonon frequencies and of the electronic transitions of TBG, the theoretical model used here does not give us the intensities of Raman peaks. For that we also need to consider the values of the matrix elements for the el–ph and electron–photon interactions. The computation of these matrix elements can clarify, through the intensity, if the peak marked as $T_a/L_a$ in Fig. 4g is, indeed, a $T_a$ or $L_a$ peak.

**Data availability**. The data that support the findings of this study are available from the corresponding author upon request.

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

## Acknowledgements

This work was partially supported by the Brazilian Institute of Science and Technology (INCT) in Carbon Nanomaterials and the Brazilian agencies Fapemig, CAPES and CNPq. H.B.R. acknowledges a CNPq scholarship and FAPESP project 2012/50259-8. The authors thank Prof. L. G. Cancado for his helpful discussions.

## Author contributions

G.S.N.E., A.R. and M.A.P. conceived the idea and designed the experiments; G.S.N.E. and H.B.R. performed the Raman map experiments; G.S.N.E. performed the resonance Raman experiments in visible range; G.S.N.E. and P.P. performed the Raman experiments in UV range; G.S.N.E., T.M. and M.P. performed the Raman experiments in IR range; G.S.N.E. and M.A.P. analysed and interpreted the experimental data; P.-W.C. prepared the TBG's samples; G.S.N.E., A.C.G. and L.C.C. prepared the graphene/h-BN heterostructures; A.C.G. performed the AFM measurements; K.W. and T.T. prepared the h-BN crystal; G.S.N.E., M.V.O.M., P.V. and M.A.P. developed the theoretical model; M. V.O.M. and P.V. performed the theoretical calculations. G.S.N.E., M.V.O.M., P.V. and M. A.P. wrote the paper; the work was supervised by A.R., P.V. and M.A.P.; all authors discussed the results and commented on the manuscript.

## Additional information

**Competing interests:** The authors declare no competing interests.

