## [Peer Review File(PDF 261 kb) · Nature Communications]

Reviewers' comments:

Reviewer #1 (Remarks to the Author):

This is a very interesting piece of work that clarifies the origins of the extra peaks observed in Raman spectra of twisted bilayer graphene (TBG). The authors carried out extensive measurements using many laser lines and compared the data with a theoretical model to justify their conclusion. The analyses are rigorous and technically sound. Although the topic would attract the interest of a relatively small number of researchers specializing in Raman analysis of graphene, the implication and the scientific rigor would justify publication in Nature Communications.

I only have a few minor comments that the authors need to address when they prepare the final version of the manuscript.

1. Although one can refer to electrons being in one layer or another, one cannot do the same about the phonons because phonons are essentially normal vibration modes of the entire crystal. The authors say, for example, "In atomically thin heterostructures, the interaction can involve both electrons and phonons in the same layer (intralayer el-ph interaction) or in adjacent layers (interlayer el-ph interaction)." Although it is perfectly all right to mention interlayer (or intralayer) electron-phonon scattering, the expression such as 'phonons in the same layer' should be avoided.

2. Page 3, 9th line from the bottom of the right column: More explanation is needed for the following claim, '... in the case of TBG, the back-scattering is provided by a periodic potential of the Moire pattern.' Unlike the disorder-induced D and D' bands, the periodic potential of the Moire pattern has a definite orientation. As such, it should be checked whether the momentum conservation would work in this case.

3. Caption of Fig. 3(c): 'green and red open circles' should be 'green and red open squares'.

4. The authors refer to Raman measurements with excitation energies of 1.49, 1.70, and 1.82 eV as 'IR Raman'. However, 1.70 and 1.82 eV photons are by no means infrared.

5. There is no description of the sample preparation. Although the details have been published elsewhere, a summary of the sample preparation methods should be given in the Experimental Methods section.

6. There are some grammatical errors in the text. A thorough proofreading is recommended.

Reviewer #2 (Remarks to the Author):

The authors report a meticulous study on intralayer and interlayer electron-phonon interactions in twisted bilayer graphene through Raman spectroscopy measurements. They observed additional Raman peaks in twisted bilayer graphene and carefully checked its behaviors in terms of the excitation laser energy and the twisting angle. DFT calculations were done to compare the experimental results with theoretical model. The conclusion is reasonable and the results are meaningful to the research community. However, there are still some concerns which need to be further clarified. I suggest that a major revision is needed.

My comments and concerns are as follows

(1) In page 4, "These two maximum are associated with electronic transitions close to K and M points of the graphene, and will be called E^{-} and E^{+} ." From only this sentence, it is difficult to make sense. As far as I understand, E^{-} is shown in Fig.2(b). E^{+} is not easy to

imagine. To make it clarified, I suggest to show a schematics for $E^{\{+\}}$.

(2) As the authors mentioned in page 5, all of them, L_a , L_e , T_a and T_e , should be active in any twisting angle. However, some peaks were not observed in such condition. Could you explain the reason?

(3) In a same context of comments (2), could you plot $I_{\{L_a\}}/I_{SLG}$, $I_{\{L_e\}}/I_{SLG}$, $I_{\{T_a\}}/I_{SLG}$ and $I_{\{T_e\}}/I_{SLG}$ vs. the twisting angle when the intensity is maximum in such photon energy? I think you can compare these plots to your calculations. These plots can help to confirm your theoretical model.

(4) I found minor mistakes in this manuscript. Corrections are needed.

a. In Fig.3(c) and (d), green squares should be changed to green circle.

b. In Fig. 3(c), the explanation for red and black crosses are missed.

c. In reference 18, volume and page numbers are missed.

Reviewer #3 (Remarks to the Author):

In the present work, the authors report on the observation of specific Raman peaks related to both intralayer and interlayer electron-phonon interactions in twisted graphene bilayers with different twisting angles. They also propose that the intensity of these extra peaks could provide information relative to the strength of the interaction between graphene and the substrate (h-BN).

This work is scientifically sound, original, and the scientific discussion is quite convincing. This research results in a major conceptual leap forward in the Raman study of electron-phonon interactions in graphene heterostructures. At last, the present work also proposes a novel clear convention for these extra Raman peaks (L_a , L_e , T_a , T_e) that is clarifying the previous ambiguous notations (R and R' peaks).

The only minor weak point is the absence of critical scientific background regarding the use of the simple folding of the SLG theoretical calculations to capture the physics of both intralayer and interlayer processes in twisted bilayer graphene. Some additional discussion would have been helpful to understand some observed discrepancies (i.e. Fig.4.g).

To my opinion, the present manuscript meets the guidelines for publication in Nature Communications, and I would thus strongly recommend its publication.

Reviewers' comments

Reviewer #1 (Remarks to the Author):

This is a very interesting piece of work that clarifies the origins of the extra peaks observed in Raman spectra of twisted bilayer graphene (TBG). The authors carried out extensive measurements using many laser lines and compared the data with a theoretical model to justify their conclusion. The analyses are rigorous and technically sound. Although the topic would attract the interest of a relatively small number of researchers specializing in Raman analysis of graphene, the implication and the scientific rigor would justify publication in Nature Communications.

Answer: We thank Reviewer #1 for recognizing the quality and importance of our work.

I only have a few minor comments that the authors need to address when they prepare the final version of the manuscript.

1. *Although one can refer to electrons being in one layer or another, one cannot do the same about the phonons because phonons are essentially normal vibration modes of the entire crystal. The authors say, for example, "In atomically thin heterostructures, the interaction can involve both electrons and phonons in the same layer (intralayer el-ph interaction) or in adjacent layers (interlayer el-ph interaction)." Although it is perfectly all right to mention interlayer (or intralayer) electron-phonon scattering, the expression such as 'phonons in the same layer' should be avoided.*

Answer: We agree with this comment and we have corrected the sentences in the Abstract and in the 1st paragraph of the manuscript, which is now written as.

"In atomically thin heterostructures, the interaction can involve electrons in the same layer (intralayer el-ph interaction) or in adjacent layers (interlayer el-ph interaction)."

2. *Page 3, 9th line from the bottom of the right column: More explanation is needed for the following claim, '... in the case of TBG, the back-scattering is provided by a periodic potential of the Moire pattern.' Unlike the disorder-induced D and D' bands, the periodic potential of the Moire pattern has a definite orientation. As such, it should be checked whether the momentum conservation would work in this case.*

Answer: In the paragraph starting with “In order to determine...” in page 4, we explain how the restricted density of states is calculated. In the case of the *intralayer* process, we consider scattering between electronic states \mathbf{k} and $\mathbf{k} + \mathbf{q}_M$, being \mathbf{q}_M a Moiré reciprocal lattice vector. In this condition, momentum conservation is always achieved in our model by a given state \mathbf{k} . In order to clarify this important point, we have added the following sentence in page 4.

Considering these two conditions, momentum conservation is always achieved in our model by a given pair of state \mathbf{k} and \mathbf{k}' , as shown in Figs. 2(d) and 2(e).

3. *Caption of Fig. 3(c): ‘green and red open circles’ should be ‘green and red open squares’.*

Answer: We have corrected the caption of Fig. 3.

4. *The authors refer to Raman measurements with excitation energies of 1.49, 1.70, and 1.82 eV as ‘IR Raman’. However, 1.70 and 1.82 eV photons are by no means infrared.*

Answer: This point has been corrected throughout the manuscript.

5. *There is no description of the sample preparation. Although the details have been published elsewhere, a summary of the sample preparation methods should be given in the Experimental Methods section.*

Answer: We have added the following sentence in the Experimental Methods section:

The TBG samples were grown by CVD technique using methane (99.99%) on polycrystalline Cu foils [12]. The graphene layers were first covered by a thin layer of polycarbonate, followed by etching in HCl aqueous solution to remove the Cu in the transfer process. The polycarbonate film with attached graphene was then transferred onto different substrates: a 300 nm SiO₂/Si, 90 nm SiO₂/Si and fused silica. Finally, the polycarbonate film was removed using chloroform. The gr/BN samples were prepared by mechanical exfoliation of graphene and transference to a h-BN substrate

6. *There are some grammatical errors in the text. A thorough proofreading is recommended.*

Answer: We have made a full revision of the manuscript.

Reviewer #2 (Remarks to the Author):

The authors report a meticulous study on intralayer and interlayer electron-phonon interactions in twisted bilayer graphene through Raman spectroscopy measurements. They observed additional Raman peaks in twisted bilayer graphene and carefully checked its behaviors in terms of the excitation laser energy and the twisting angle. DFT calculations were done to compare the experimental results with theoretical model. The conclusion is reasonable and the results are meaningful to the research community.

Answer: We also thank Reviewer #2 for recognizing the quality and importance of our work.

However, there are still some concerns which need to be further clarified. I suggest that a major revision is needed.

My comments and concerns are as follows.

(1) In page 4, "These two maximum are associated with electronic transitions close to K and M points of the graphene, and will be called E^{-} and E^{+} ." From only this sentence, it is difficult to make sense. As far as I understand, E^{-} is shown in Fig.2(b). E^{+} is not easy to imagine. To make it clarified, I suggest to show a schematics for E^{+} .

Answer: We thank Reviewer #2 for this suggestion. We have included an inset in Fig. 3c showing schematically the processes associated to E^{-}_a , E^{-}_e , E^{+}_a and E^{+}_e . In order to improve the comprehension of Fig 3c, we changed the color of the previous continuous black curve, that is now presented in dashed green. We added the following sentence in the manuscript:

In the caption of Fig. 3c we show the low and high energy *intralayer* and *interlayer* processes. Notice that, in the limit of small and large twisting angles, the low energy transitions involve states near the **K point of graphene, whereas the high energy transitions involve states near the **M** point.**

(2) As the authors mentioned in page 5, all of them, L_a , L_e , T_a and T_e , should be active in any twisting angle. However, some peaks were not observed in such condition. Could you explain the reason?

Answer: This issue has briefly discussed in the paragraph in the right column of page 5 starting as: "In principle, the interlayer electron-phonon scattering process can also activate LO phonons...". As

suggested by Reviewer #2, we have reviewed and extended this paragraph in order to clearly address this point. In fact, in our Raman experiments in TBG using visible light, we have not observed the L_e and T_a peaks. The absence of L_e can be only due to the fact that this peak might appear in spectra where the G band is hugely enhanced. Since the L_e peak is expected to appear very close to the G band using visible photons, it is possibly masked by the G band enhancement. The absence of T_a is not yet well understood. It is possibly due to the very small Raman cross-section of TO phonons in the *intralayer* process using visible photons. However, a complete calculation of the cross-section will be needed to explain these results. The new version of the paragraph is now:

In principle, the intralayer and interlayer el-ph processes can also activate TO and LO phonons, respectively, and give rise to T_a and L_e peaks for samples with intermediate angles. However, T_a and L_e were not observed in our multiple excitation experiments using visible photons. The lack of observation of L_e can be due to the huge enhancement of the G band. Since the position of the L_e peak is very close to the G band position in samples with intermediate twisting angles, it is possibly masked by the G band enhancement. For samples with θ around 10° and measured using the 1.96 eV laser line, Campos- Delgado et al. [9] reported the observation of a peak at 1622 cm^{-1} , that might be assigned to the L_e . The absence of T_a can be ascribed to the very weak cross-section of TO phonons activated by the *intralayer* process. Electron-phonon matrix elements calculations would be necessary to better clarify this issue.

(3) In a same context of comments (2), could you plot I_{L_a}/I_{SLG} , I_{L_e}/I_{SLG} , I_{T_a}/I_{SLG} and I_{T_e}/I_{SLG} vs. the twisting angle when the intensity is maximum in such photon energy? I think you can compare these plots to your calculations. These plots can help to confirm your theoretical model.

Answer: As suggested by Reviewer 2, we have included a new figure in the Supplementary Material (SM) with the plots I_{L_a}/I_{SLG} and I_{T_e}/I_{SLG} vs. the twisting angle. In this work, we only calculated the frequencies and laser energies where the new peaks should appear, but not the intensities of each peak. The calculation of Raman intensities is more challenging, since we need to consider the values of the matrix elements for the electron-phonon and electron-photon interactions. Future calculations will be needed to describe the Raman intensities of the extra peaks. We have added the following sentence in the SM in order to explain the new Fig. 8 of the SM.

Figure 8 shows the relative intensities of T_a , T_e , L_a and L_e as a function of the twisting angle θ . Notice that they increase with decreasing values of θ .

Departamento de Física
Instituto de Ciências Exatas
Universidade Federal de Minas Gerais
Raman Spectroscopy Laboratory
C.P 702, Belo Horizonte, MG 30123-970, Brazil

Marcos A. Pimenta
Professor of Physics
+55-31-3409-6622
E-mail: mpimenta@fisica.ufmg.br

In the paragraph of the manuscript which starts as "The calculated results presented in Fig.3c show that ..." we have added the following sentence:

This effect is shown in the supplementary Figure 8 that plots the relative intensities of T_a , T_e , L_a and L_e as a function of the twisting angle θ .

(4) I found minor mistakes in this manuscript. Corrections are needed.
a. In Fig.3(c) and (d), green squares should be changed to green circle.

Answer: We reviewed and fixed the errors.

b. In Fig. 3(c), the explanation for red and black crosses are missed.

Answer: This explanation has been corrected.

c. In reference 18, volume and page numbers are missed.

Answer: This reference has been corrected.

Reviewer #3 (Remarks to the Author)

In the present work, the authors report on the observation of specific Raman peaks related to both intralayer and interlayer electron-phonon interactions in twisted graphene bilayers with different twisting angles. They also propose that the intensity of these extra peaks could provide information relative to the strength of the interaction between graphene and the substrate (h-BN).

This work is scientifically sound, original, and the scientific discussion is quite convincing. This research results in a major conceptual leap forward in the Raman study of electron-phonon interactions in graphene heterostructures. At last, the present work also proposes a novel clear convention for these extra Raman peaks (L_a , L_e , T_a , T_e) that is clarifying the previous ambiguous notations (R and R' peaks).

Answer: We also thank Reviewer #3 for pointing out the importance of our work.

The only minor weak point is the absence of critical scientific background regarding the use of the simple folding of the SLG theoretical calculations to capture the physics of both intralayer and interlayer processes in twisted bilayer graphene. Some additional discussion would have been helpful to understand some observed discrepancies (i.e. Fig.4.g).

Answer: We used the folding approach since this method has been used in the literature to describe the electronic transitions in TBG with relative success [15-17]. Concerning the phonon dispersion, it was shown by Cocemasov et al. (Ref. 24 in the new version of the manuscript) that the frequencies of the TBG branches are not significantly different to those of the single layer graphite in the range considered in our work. In order to clarify this point, we have included this new reference in the manuscript and the following paragraph in the end of the Theoretical Methods section.

“Concerning the electronic properties, it has been shown that interlayer interactions perturb the bands from each layer producing mini-gaps in the conduction and valence band of TBG with same energy. However, the allowed transitions do not depend on the size of the minigap, and therefore the folding approach is supposed to give good results concerning the Raman processes investigated here [17]. As far as phonon properties are concerned, Cocemasov et al. [24] have shown that the weak van der Waals interactions between layers in TBG do not alter significantly the frequencies of the branches in the range considered here. Although the folding method provides a good description of the phonon frequencies and of the electronic transitions of TBG, the theoretical model used here does not give us the intensities of Raman peaks. For that, we also need to consider the values of the matrix elements for the electron-phonon and electron-photon interactions. The computation of these matrix elements can clarify the origin of the T_a/L_a peak in Fig. 4g.”

Departamento de Física

Instituto de Ciências Exatas
Universidade Federal de Minas Gerais
Raman Spectroscopy Laboratory
C.P 702, Belo Horizonte, MG 30123-970, Brazil

Marcos A. Pimenta

Professor of Physics
+55-31-3409-6622
E-mail: mpimenta@fisica.ufmg.br

To my opinion, the present manuscript meets the guidelines for publication in Nature Communications, and I would thus strongly recommend its publication

REVIEWERS' COMMENTS:

Reviewer #1 (Remarks to the Author):

The authors have addressed the comments and concerns of the referees adequately. The only issue that needs to be addressed during the production process is the visibility of the figures. The inset of Figure 3c, in particular, is almost impossible to see due to very small size fonts.

Reviewer #2 (Remarks to the Author):

The authors did corrections on the manuscript as following comments of reviewers. I recommend this work to be published on Nature Communications.

Reviewer #3 (Remarks to the Author):

The points raised by the authors concerning the absence of critical scientific background regarding the use of the simple folding of the SLG theoretical calculations to capture the physics of both intralayer and interlayer processes in twisted bilayer graphene have been satisfactorily addressed.

Consequently, I would recommend its publication in Nature Communications.